# Structure of the human frataxin-bound iron-sulfur cluster assembly complex provides insight into its activation mechanism

Nicholas G. Fox[1,4,6], Xiaodi Yu [2,5,6], Xidong Feng[2], Henry J. Bailey[1], Alain Martelli[3], Joseph F. Nabhan[3], Claire Strain-Damerell[1], Christine Bulawa[3], Wyatt W. Yue [1] & Seungil Han [2]

The core machinery for de novo biosynthesis of iron-sulfur clusters (ISC), located in the mitochondria matrix, is a five-protein complex containing the cysteine desulfurase NFS1 that is activated by frataxin (FXN), scaffold protein ISCU, accessory protein ISD11, and acyl-carrier protein ACP. Deficiency in FXN leads to the loss-of-function neurodegenerative disorder Friedreich's ataxia (FRDA). Here the 3.2 Å resolution cryo-electron microscopy structure of the FXN-bound active human complex, containing two copies of the NFS1-ISD11-ACP-ISCU-FXN hetero-pentamer, delineates the interactions of FXN with other component proteins of the complex. FXN binds at the interface of two NFS1 and one ISCU subunits, modifying the local environment of a bound zinc ion that would otherwise inhibit NFS1 activity in complexes without FXN. Our structure reveals how FXN facilitates ISC production through stabilizing key loop conformations of NFS1 and ISCU at the protein–protein interfaces, and suggests how FRDA clinical mutations affect complex formation and FXN activation.

[1] Structural Genomics Consortium, Nuffield Department of Clinical Medicine, University of Oxford, Oxford OX3 7DQ, UK. [2] Discovery Sciences, Worldwide Research and Development, Pfizer Inc., Eastern Point Road, Groton, CT 06340, USA. [3] Rare Disease Research Unit, Worldwide Research and Development, Pfizer Inc., 610 Main Street, Cambridge, MA 02139, USA. [4] Present address: Merck & Co, 2000 Galloping Hill Rd, Kenilworth, NJ 07033, USA. [5] Present address: SMPS, Janssen Research and Development, 1400 McKean Rd, Spring House, PA 19477, USA. [6] These authors contributed equally: Nicholas G. Fox, Xiaodi Yu. Correspondence and requests for materials should be addressed to W.W.Y. (email: wyatt.yue@sgc.ox.ac.uk) or to S.H. (email: seungil.han@pfizer.com)

Iron-sulfur clusters (ISC) are inorganic cofactors essential in all life forms with common roles in electron transfer, radical generation, and structural support[1]. In eukaryotes, the de novo ISC assembly machinery is located in the mitochondrial matrix and requires a core complex comprising the proteins NFS1, ISD11, ACP, and ISCU (SDAU)[1,2]. The NFS1 cysteine desulfurase facilitates a pyridoxal 5′ phosphate (PLP) cofactor to generate the sulfane sulfur from L-cysteine, and deliver it to the ISCU scaffold protein[3,4]. The accessory protein ISD11/LYRM4 is unique in eukaryotes, and was shown to stabilize NFS1 and interact directly with the acyl carrier protein ACP/NDUFAB1[5,6]. ISCU utilizes three of its conserved cysteine residues (Cys69, Cys95, Cys138) to combine the sulfane sulfur from NFS1 with an iron source, resulting in ISC formation. ISCU then exploits the highly conserved LLPVK motif for interaction with the chaperones, such as GRP75/HSCB[7], for the downstream delivery to apo-targets. Whereas electrons required for ISC assembly most likely involves mitochondrial ferredoxin/ferredoxin reductase combination[8], the iron source remains unclear.

An intronic GAA repeat of *FXN* gene, resulting in deficiency of the frataxin (FXN) protein, causes autosomal recessive Friedreich's ataxia (FRDA)[9]. The in vivo loss of FXN results in ISC deficiency and iron accumulation in the mitochondria, rendering FRDA a fatal and debilitating condition. Several roles of FXN with respect to ISC biosynthesis, including that of an iron source[10], have been speculated. The emerging role is that FXN acts as an allosteric regulator of ISC assembly, and stimulates NFS1 activity by binding the SDAU complex to form the five-way active SDAUF complex[11–13]. ISC biosynthesis must be heavily regulated to avoid iron and sulfur toxicity in the cell, rendering FXN essential in eukaryotes. $Zn^{2+}$ ion has been found to bind ISCU and completely inhibit the SDAU complex in vitro although its activity is restored by addition of FXN[14], and the in vivo relevance of the $Zn^{2+}$ effect remains to be determined. Recently, crystal structures of SDA/SDAU/SDAU-$Zn^{2+}$ complexes without the key component FXN have been published[5,15], which attributed the zinc inhibition to the sequestration of key NFS1 catalytic residue Cys381, but could not serve as template to understand the molecular roles of FXN activator. To this end, we pursue structure determination of the SDAUF complex, coupled with FXN binding studies, to decipher the FXN-mediated activation mechanism.

## Results and discussion

**Cryo-electron microscopy of recombinant SDAUF complex.** FXN binding to the SDAU complex is dynamic, yielding low-μM dissociation constants ($K_d$) by bio-layer interferometry (BLI) (Supplementary Fig. 1a-c), hence presenting challenges to isolate the SDAUF complex intact with all 5 components in proper stoichiometry. Our several attempts to generate the SDAUF complex by reconstitution of individually expressed components (Supplementary Fig. 2a, b) did not fully incorporate FXN. To remedy this, we co-expressed in *E. coli* a plasmid containing His$_6$-ISD11-NFS1-ISCU, with a plasmid containing His$_6$-FXN (Supplementary Fig. 2c). This produced excess FXN, shifting equilibrium towards formation of a stable and active SDAUF complex comprising human SDUF co-purified with *E. coli* ACP (ACP$_{ec}$). We attempted to generate the 5-way all-human complex, without ACP$_{ec}$, by inserting human ACP (NDUFAB1) into the second site of the vector containing His$_6$-FXN. Upon co-expression with the plasmid containing His$_6$-ISD11-NFS1-ISCU, we observed a heterogeneous complex containing an approximately equimolar mixture of the desired human ACP and contaminating ACP$_{ec}$ (Supplementary Fig. 2d). Based on previous reports[5], and the functional conservation of human and *E. coli* ACP, we continued

our experiments with a homogenous complex containing ACP$_{ec}$ and human SDUF (hereafter SDAUF). The as-isolated five-way complex could still be inhibited by $Zn^{2+}$ (Supplementary Fig. 3, SDAUF), as shown previously with the four-way complex[14] (Supplementary Fig. 3, SDAU), due to the dissociation equilibrium of FXN. Addition of more purified ISCU to the five-way complex further exacerbated the $Zn^{2+}$ inhibition (Supplementary Fig. 3, SDAUF + U), while $Zn^{2+}$ inhibition was fully reversed upon further FXN supplementation to the five-way complex (Supplementary Fig. 3, SDAUF + F and SDAUF + U + F), explaining how excess FXN is needed to maintain its bound state within the five-way complex for the activation.

We determined the single-particle cryo-electron microscopy (cryo-EM) structure of the 186-kDa SDAUF complex to 3.2 Å resolution (Supplementary Fig. 4 and Supplementary Table 1), allowing model building of entire complex components, with unambiguous placement of cofactor pyridoxal 5′-phosphate covalently-linked to NFS1 Lys258, a $Zn^{2+}$ ion in each ISCU, and a 4′-phosphopantetheine (4′-PP) acyl-chain attached to ACP$_{ec}$ Ser37 (Supplementary Fig. 5). LC-MS revealed a mixture of ACP components with different acyl-chains[16], and further top-down MS[3] enabled the detailed structure elucidation on these acyl-chains through accurate mass measurements of both parent and fragment ions of the 4′-PP acyl chains which were readily ejected from ACP in the gas phase and interrogated further by tandem MS (Supplementary Fig. 6 and Supplementary Table 2). The relative abundances of the ACP components were estimated on the basis of extracted ion chromatograms in the LC-MS measurements, showing that ACP with longer acyl-chains are clearly the dominant species (Supplementary Table 2).

**Overall architecture of the SDAUF complex.** Our human SDAUF-$Zn^{2+}$ structure, the only FXN-bound complex structure to date from any organism, is a symmetric heterodecamer comprising 2 copies each of the five proteins i.e. (NFS1)$_2$(ISD11)$_2$(ACP$_{ec}$)$_2$(ISCU-$Zn^{2+}$)$_2$(FXN)$_2$. Structurally it constitutes a (NFS1-ISD11-ACP$_{ec}$)$_2$ homodimeric core, with one ISCU appended to each long end of the core, and one FXN fitted into the cavity next to each ISCU (Fig. 1a, b). This architecture agrees with small-angle x-ray scattering (SAXS) analysis (Fig. 1c, d and Supplementary Fig. 7a, b), whereby a theoretical SAXS profile back-calculated from our SDAUF-$Zn^{2+}$ cryo-EM structure shows a good fit to experimental scattering data ($\chi^2 = 1.98$). While our five-way complex superimposes well with four-way SDAU/SDAU-$Zn^{2+}$ structures from Boniecki et al.[5] within the (NFS1-ISD11-ACP$_{ec}$)$_2$ core (rmsd 0.6 Å, over 756 $C^\alpha$ atoms), there is significant displacement of ISCU, up to 2.0 Å away from the core, in our FXN-bound complex (Supplementary Fig. 7c).

FXN occupies a cavity at the interface of both NFS1 and one ISCU subunits (Fig. 1e and Supplementary Fig. 8a), burying ~ 1345 Å$^2$ of FXN accessible surface. No direct FXN-ISD11 interaction is observed, contrasting previous predictions with oligomeric FXN[17,18]. A key feature of FXN binding is its simultaneous interactions with both NFS1 protomers of the complex (Fig. 2a), which definitively supports previous predictions from crosslinking, SAXS and NMR studies[5,19,20]. Importantly, this requires a homodimeric arrangement of NFS1 within the complex, consistent with the SDAU conformation observed by Boniecki et al.[5], while incompatible with the SDA$_{ec}$ crystal structure from Cory et al. whereby an extensive NFS1 homodimer interface was not observed[15]. The SDA conformation from Cory et al., not observed in all cysteine desulfurase structures published to date, would therefore not be conducive to the FXN activation mechanism, and could possibly instead depict a FXN-independent function.

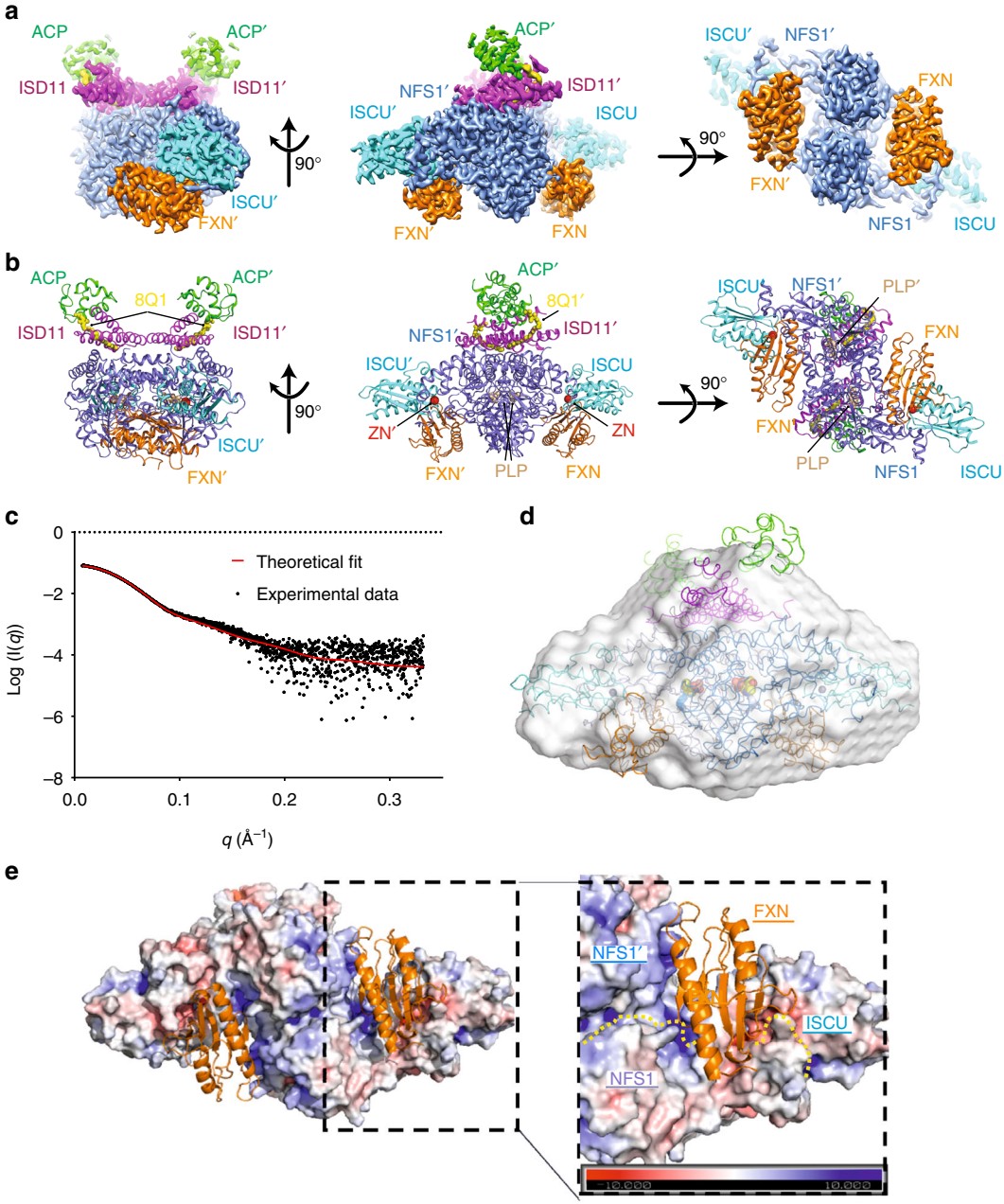

**Fig. 1** SDAUF-Zn$^{2+}$ structure and FXN-NFS1 interactions. **a**. Cryo-EM density of SDAUF-Zn$^{2+}$ structure (NFS1/NFS1' slate, ISD11/ISD11' magenta, ACP/ACP' light green, ISCU/ISCU', Cyan, FXN/FXN' orange). **b**. Cartoon representation of SDAUF-Zn$^{2+}$ complex. 4'-phosphopantetheine acyl chain (8Q1, yellow), Zn$^{2+}$ ion (ZN, red) and pyridoxal 5'-phosphate (PLP, wheat) are shown as sticks/spheres. **c**. Scattering data from a sample of SDAUF used in cryo-EM was collected (black points) and fit to the theoretical SAXS profile back-calculated from the SDAUF-Zn$^{2+}$ cryo-EM structure (red line) with $\chi^2 = 1.98$. **d**. The ab initio envelope calculated from SAXS data for the SDAUF sample was superimposed with the cryo-EM structure **e**. In the SDAUF complex, each FXN binds to a cavity formed by interface of NFS1 homodimer and one ISCU. FXN is shown as orange cartoon, while other subunits of the complex are shown as surface representation (colored by electrostatic potential). Yellow dotted lines denote protein boundaries. Source data for panel c are provided as a Source Data file

**FXN Interactions with NFS1 dimer interface and C-terminus**. FXN interacts with one NFS1 protomer via potential salt-bridges, and with the other NFS1 protomer through van der Waals contacts (Fig. 2a). The salt-bridge interface, highly conserved across orthologues (Supplementary Fig. 9), involves an acidic ridge of FXN[21] (end of α1, loop α1-β1, start of β1) and a positively-charged Arg-rich patch (Arg272-Arg277) on one NFS1 protomer (Figs. 1d, 2a and Supplementary Fig. 8a, b). For example, FXN Asp124 can potentially form salt-bridges with NFS1 Arg289, while carbonyl backbones of FXN residues Glu121 and Tyr123

form potential H-bonds with NFS1 Arg119 and Arg272 (Fig. 2a). The FXN(D124A) variant, aimed at abolishing these salt-bridges, binds SDAU with 4-fold weakened $K_d$ compared to wild-type (WT)(Table 1 and Supplementary Fig. 1d). FXN contacts the other NFS1 protomer, and ISCU, using the β-sheet (β1-β5) surface (Fig. 2a–d). In this interface, FXN directly contacts the NFS1 loop containing the catalytic residue Cys381 (Cys-loop) (Fig. 2a), via hydrophobic interaction between FXN Trp155 and NFS1 Leu386, and potential H-bond between FXN Asn146 and NFS1 Ala384 carbonyl backbone. The clinical variant FXN(N146K)

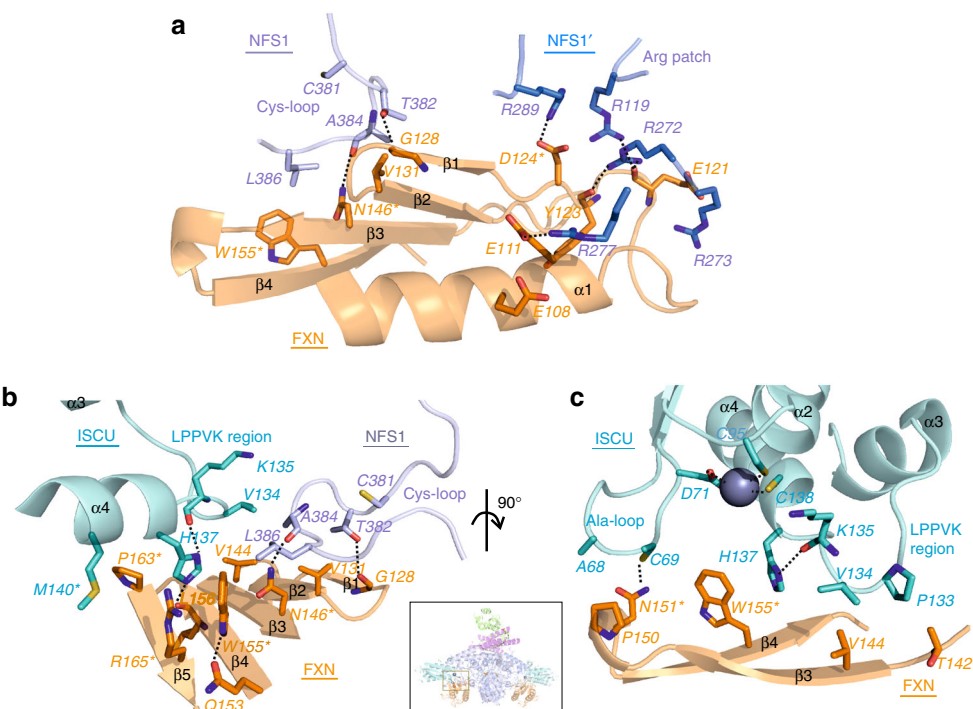

**Fig. 2** FXN-ISCU interactions and FXN mutagenesis. **a**. Interactions of FXN (orange) with both NFS1 subunits (NFS1, light slate; NFS1', blue). Residues studied by site-directed mutagenesis are asterisked. Dashed lines denote potential hydrogen bonds. **b**. Interface of FXN β-sheet with ISCU LPPVK-region and NFS1 Cys-loop. **c**. Interface of FXN with ISCU Ala-loop, LPPVK-region, and $Zn^{2+}$ ion (sphere). Inset shows viewpoints of panels b and c within SDAUF-$Zn^{2+}$ complex

**Table 1 Dissociation constant ($K_d$) and melting temperature ($T_m$) of FXN variants**

|  | BLI | | DSF | |
| --- | --- | --- | --- | --- |
|  | $K_d$ (uM) | Error | $T_m$ (ºC) | Error |
| WT | 3.32 | 0.94 | 66.45 | 0.23 |
| D124A | 14.36 | 1.53 | 65.33 | 0.70 |
| N146K | 177.2 | 43.8 | 72.51 | 0.86 |
| N151A | 9.45 | 2.45 | 66.05 | 2.33 |
| W155R | 253.3 | 67.3 | 60.85 | 0.60 |
| P163G | 38.45 | 7.16 | 66.34 | 1.19 |
| R165C | 134.2 | 8.7 | 68.96 | 1.16 |

exhibited 50-fold weakened $K_d$ towards SDAU (Table 1 and Supplementary Fig. 1d). Beyond the two extensive interfaces with FXN, NFS1 further contributes its C-terminal 20 aa (Ser437-His457) to wrap around the ISCU surface, with terminal residues Gln456-His457 anchored by FXN Asn151, Tyr175 and His177 via potential H-bonds (Supplementary Fig. 10). FXN(N151A) variant binds SDAU with 3-fold weakened $K_d$ (Table 1 and Supplementary Fig. 1d). Therefore, NFS1, via two extensive interfaces and C-terminus, anchors FXN to interact with ISCU. This likely explains why FXN alone cannot bind ISCU without NFS1[11,13,22].

**FXN binds to two key regions on ISCU.** One ISCU-FXN interface is through the conserved ISCU Ala-loop (Ala66-Asp71), containing the conserved Cys69 that is required for ISC biosynthesis and interacts with FXN Asn151, as well as the $Zn^{2+}$ coordinating ligand Asp71. This interaction (Fig. 2c), which may account for the weaker binding caused by the FXN(N151A) variant (Table 1), is mediated by significant changes of the ISCU Ala-loop conformation in SDAUF as compared with SDAU-$Zn^{2+}$, and

zinc-free SDAU structures (rmsd ~ 6 Å and ~ 2 Å respectively, over 24 superposed Cα atoms) (Fig. 3a). The other, more predominant ISCU-FXN interface is through the conserved ISCU $L_{131}PPVKLHCSM_{140}$ sequence motif (Fig. 3b). This region (Fig. 2b, c), connecting ISCU helices α3 and α4, contains: the $L_{131}PPVK_{135}$ sequences recognized by the GRP75/HSCB chaperones for downstream ISC delivery[7], Cys138 the proposed sulfur acceptor for the NFS1 sulfane[23], and Met140, a residue reportedly determining if ISC biosynthesis is FXN-dependent (as in eukaryotes) or FXN-independent (prokaryotes)[24].

Previous structures of zinc-bound ISCU reveal a helical conformation for the $L_{131}PPVK_{135}$ region[5]. In our structure, the displacement of ISCU caused by FXN (Supplementary Fig. 7c) is associated with the ISCU $L_{131}PPVK_{135}$ helix becoming loosened and more flexible, allowing His137 to pack against invariant FXN Trp155 (Fig. 2b, c). ISCU Pro133 and Val134 also pack against FXN Thr142 and Val144, respectively (Fig. 2c), in turn stabilizing NFS1 Cys-loop through interactions with NFS1 Ala384 and Leu386 (Fig. 2b). Considering the key role of FXN Trp155 in mediating ISCU and NFS1 conformations within the complex, the FXN(W155R) clinical variant of FRDA has drastically weakened $K_d$ for SDAU by > 75-fold, compared to FXN(WT) (Table 1 and Supplementary Fig. 1d). This observation is consistent with FXN(W155R) being an early-onset and one of the most clinically severe point variant for FRDA[25]. Trp155 is further held in place by FXN Arg165 (pi-stacking interaction) and Gln153 (potential H-bond) (Fig. 2b). The clinical variant FXN (R165C) has weakened $K_d$ by 40-fold (Table 1 and Supplementary Fig. 1d). Our binding studies with the FXN(N146K) (previous section) and FXN(R165C) variants are consistent with their relatively milder clinical phenotypes[26,27], considering that these two amino acids are engaged in less interactions than Trp155.

Substitution of the ISCU Met140 residue in yeast Isu, to the amino acids (Ile, Leu, or Val) observed at the equivalent

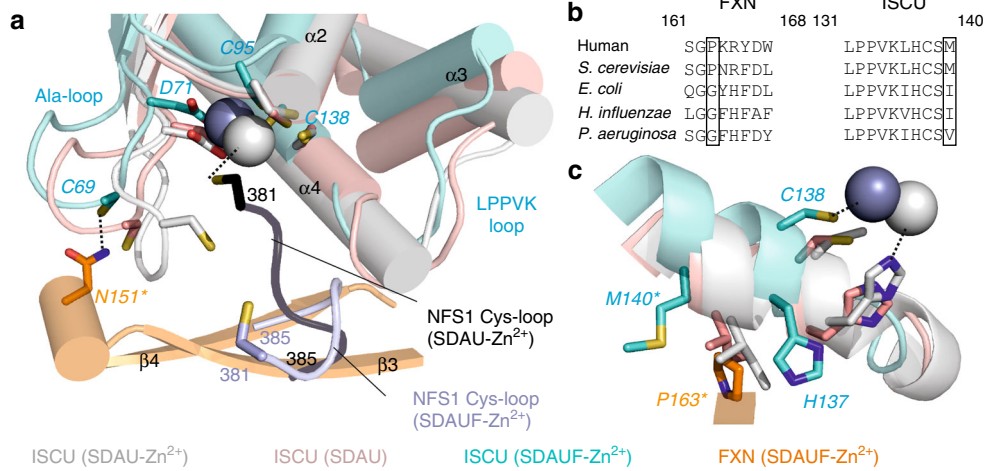

**Fig. 3** Comparing structures with and without FXN. **a**. Same view of SDAUF-Zn²⁺ as Fig. 2c, superimposed with ISCU subunits from SDAU-Zn²⁺ (5WLW) and SDAU (5WKP) structures. NFS1 Cys-loops from SDAUF-Zn²⁺ and SDAU-Zn²⁺ are shown. **b**. Sequence alignment of FXN region containing Pro163 (human numbering) and the ISCU $L_{131}$PPVKLHCSM$_{140}$ loop. Sequences shown include FXN and ISCU orthologues from human (Uniprot ID Q16595 and Q9H1K1, respectively), *S. cerevisiae* (Q07540, Q03020), *E. coli* (P27838, P0ACD4), *H. influenzae* (P71358, Q57074) and *P. aeruginosa* (Q9HTS5, Q9HXI9) **c**. ISCU $L_{131}$PPVKLHCSM$_{140}$ loop from the superimposed structures, adjacent to FXN Pro163 (orange). Human ISCU encodes Met at residue 140 (cyan stick), while in SDAU-Zn²⁺ and SDAU structures Met was substituted to Ile (white and pink sticks respectively). In both panels a and c, FXN mutagenesis residues and ISCU Met140 are asterisked, and Zn²⁺ ions are shown as spheres. Dashed lines denote potential hydrogen bonds or Zn²⁺ coordination

prokaryotic position (Fig. 3b), obviated the need for Yfh1 (FXN equivalent) and reversed ΔYfh1 phenotype[28]. Our structure shows that ISCU Met140 packs against FXN Pro163 (Fig. 3c), and a M140I substitution (present in the SDAU/SDAU-Zn²⁺ structures)[5] sterically clashes with Pro163, unless the 163 position adopts the bacterial equivalent amino acid, Gly (*E. coli* Gly68; Fig. 3b). Our structure hence illustrates the evolutionarily distinct Met:Pro pairing in eukaryotes and Ile:Gly in prokaryotes. FXN (P163G) binds SDAU 10-fold weaker than FXN(WT) (Table 1 and Supplementary Fig. 1d).

**FXN modifies the Zn²⁺ environment within the complex.** FXN binding to SDAU also influences the ISCU Zn²⁺ environment and NFS1 Cys-loop. In the reported SDAU-Zn²⁺ structure, ISCU Asp71, Cys95 and His137, and NFS1 Cys381 (from Cys-loop) form the Zn²⁺ ligation[5]. This structure explained the Zn²⁺-dependent inhibition of NFS1 activity in vitro, due to sequestration of the catalytic Cys381 away from turning over the substrate cysteine. In our SDAUF-Zn²⁺ structure, zinc is ligated by ISCU Asp71, Cys95 and Cys138 (Fig. 2c). This rearranged metal coordination frees up ISCU His137 (now 3.7 Å away from Zn²⁺) to interact with FXN Trp155, Leu156 carbonyl backbone, Pro163, and ISCU Lys135 carbonyl backbone. Importantly, NFS1 Cys381 is also freed from Zn²⁺ ligation, now available for sulfur transfer (Fig. 3a). Our structure captured a previously unobserved conformation of NFS1 Cys-loop that positioned Cys381 approximately halfway between the NFS1 active site and conserved ISCU Cys residues, as part of a loop trajectory of 27 Å that could take place during ISC assembly (Fig. 4). Supported by the activity assay (Supplementary Fig. 3), our data therefore reveals how FXN unlocks the zinc inhibition of SDAU complex, to activate NFS1 into a conformation that is now poised for its incoming substrate cysteine[14]. While the observed Zn²⁺ could mimic the ISC in aerobic in vitro experiments, it remains to be determined whether Zn²⁺ plays a role in vivo.

To conclude, our structure elucidates how FXN binding to SDAU complex causes significant conformational changes to ISCU, which unlocks the zinc inhibition and primes its key regions (Ala-loop and LPPVK region) to facilitate mobility of

NFS1 Cys-loop for sulfide formation and transfer during ISC assembly (Fig. 4). This work provides the framework for future mechanistic studies on the dynamics of SDAUF complex during next steps in ISC biosynthesis. For example, the LPPVK region of ISCU, proposed to be important for downstream chaperone binding, is buried at the FXN interface in our structure, raising the question whether FXN would be released from the complex to make way for binding ferredoxin (FDX) or the GRP75/HSCB chaperones[29]. If so, the complex would cycle between the electron donation (mediated by FDX) and sulfur transfer (mediated by FXN) steps in ISC biosynthesis. Since only reduced FDX is proposed to interact with the complex[22], it is possible that once electron donation is complete, oxidized FDX would dismount from complex and make room for another reduced FDX or FXN. Future studies are warranted to determine the dependency of FDX binding.

This work also represents one of very few reported cryo-EM structures of < 200 kDa and < 3.5 Å resolution for both membrane and soluble proteins. Our ability to visualize ligands and cofactors in such depth shows the advancement of modern cryo-EM for structure determination. We expect more examples to follow as the technique is applied to a broad range of clinically-relevant targets that remain intractable for X-ray crystallography.

## Methods

**Protein expression and purification**. For bi-cistronic co-expression of NFS1-ISD11, a DNA fragment encoding His-tagged ISD11 and non-tagged NFS1 (Δ1-55), interspersed by an in-frame ribosomal binding site, was sub-cloned into the pNIC28-Bsa4 vector (GenBank ID: EF198106). For tri-cistronic co-expression of NFS1-ISD11-ISCU, a DNA fragment encoding His-tagged ISD11, non-tagged NFS1 (Δ1-55) and non-tagged ISCU (Δ1-34), interspersed by two in-frame ribosomal binding sites, was sub-cloned into the pNIC28-Bsa4 vector. Construct encoding human FXN (Δ1-80) was sub-cloned into the pCDF-LIC-Bsa4 vector, and construct encoding human ISCU (Δ1-34) isoform 1 was sub-cloned into the pNIC28-Bsa4 vector. Primers used in this study are listed in Supplementary Table 3.

For recombinant expression, *E. coli* BL21(DE3)-R3-pRARE2 cells transformed with the above plasmids were cultured in Terrific Broth, induced with 0.1 mM isopropyl β-D-1-thiogalactopyranoside (IPTG), and incubated for 16 h at 18 °C. Harvested cell pellets were resuspended in binding buffer containing 50 mM HEPES pH 7.5, 500 mM NaCl, 20 mM Imidazole, 5% glycerol, 2 mM TCEP and EDTA-free protease inhibitor (Merck). Resuspended cells were lysed by sonication

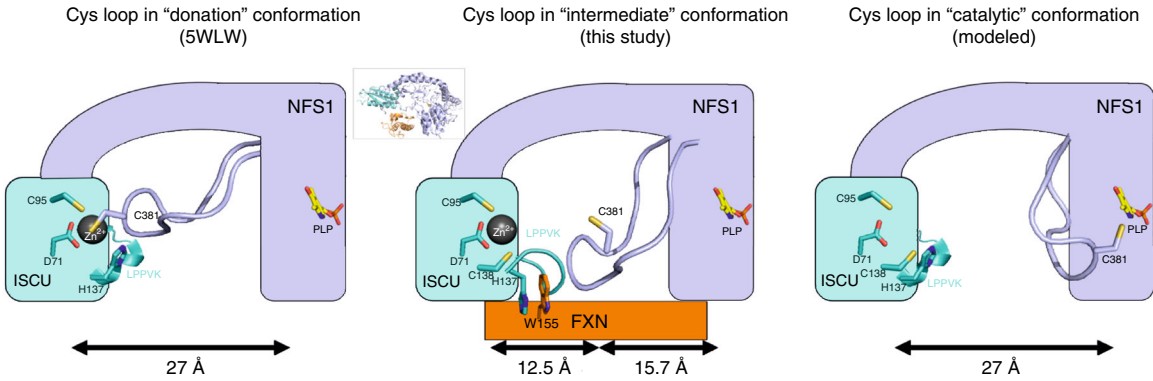

**Fig. 4** Proposed mechanistic features of FXN activation on NFS1 and ISCU. Schematic of the NFS1 Cys-loop trajectory, mediated by FXN-induced conformational changes on ISCU in the LPPVK region. Subunits of NFS1, ISCU and FXN are colored violet, cyan and orange respectively. Key residues discussed in the text are shown as sticks. The Cys-loop from NFS1 is shown as coil, and the LPPVK region Inset shows the orientation of complex from which the cartoon representation is derived

and clarified by centrifugation. To purify samples containing NFS1, binding buffer was further supplemented with 150 μM pyridoxal 5′-phosphate (PLP) prior to lysis.

To purify proteins for enzymatic and biophysical studies (SDA, SDAU, SDAUF, ISCU and FXN), centrifuged supernatants were incubated with 2.5 mL Ni Sepharose 6 fast flow resin (GE Healthcare), washed with binding buffer containing 40 mM Imidazole, and eluted with binding buffer containing 250 mM Imidazole. Fractions containing target proteins were pooled, added with 10 mM DTT (for samples containing ISCU or NFS1), and applied to size exclusion column (Superdex S75 for ISCU and FXN, Superdex S200 for NFS1-ISD11 complex; GE Healthcare) pre-equilibrated with gel filtration buffer (50 mM HEPES pH 7.5, 200 mM NaCl, 5% glycerol, and 2 mM TCEP). Peak fractions were pooled for His-tag removal by treatment with His-TEV protease, and re-purified by passing onto Ni Sepharose 6 fast flow resin to remove residual His-TEV protease and cleaved His-tag. Purified target proteins were then buffer-exchanged into gel filtration buffer. Recombinantly expressed NFS1-ISD11 (from bi-cistronic plasmid) and NFS1-ISD11-ISCU (from tri-cistronic plasmid) complexes co-purified with *E. coli* ACP, resulting in NFS1-ISD11-ACP (SDA) and NFS1-ISD11-ACP-ISCU (SDAU) complexes, respectively.

The sample of the five-way complex (SDAUF) for cryo-EM was generated by co-transformation and expression of the tri-cistronic plasmid (encoding His-ISD11, untagged NFS1, untagged ISCU) with the plasmid encoding FXN. Purification was similar to the above protocol, but with chromatography steps in the order of Ni-affinity, dialysis in the presence of TEV protease, reverse Ni-affinity, and followed by size exclusion (Superdex S200) in cryo-EM buffer (20 mM HEPES pH 7.5, 100 mM NaCl, and 2 mM TCEP). Fractions containing SDAUF were combined with an additional 3-fold excess of FXN.

FXN variants were constructed using the Q5® Site-Directed Mutagenesis Kit (NEB) and confirmed by sequencing of plasmid DNA and intact mass spectrometry of purified proteins. Cells transformed with the above plasmids were grown in 100 mL auto-induction Terrific Broth at 37 °C for 5 h and then 20 °C for 2 days. Cell pellets were resuspended in Lysis Buffer (50 mM HEPES, pH 7.5, 500 mM NaCl, 5 % Glycerol, 2 mM TCEP, 1 μg/mL Benzonase, 1:1000 EDTA-free protease inhibitor (Merck), and 5 mg/mL lysozyme), left at room temperature for 30 min and then added 1 mL/g cells 10 % Triton X-100 and frozen for > 1 h. Purification was similar to above but gel filtration was replaced with a PD-10 desalting column. All variants and wild-type were checked for stability using differential scanning fluorimetry and values recorded in Fig. 2e.

**Differential scanning fluorimetry**. DSF was performed in a 96-well plate using an Mx3005p RT-PCR machine (Stratagene). Each well (20 μl) consisted of protein (2 μM in buffer containing 50 mM HEPES, pH 7.5, 250 mM NaCl, 5% glycerol, and 2 mM TCEP), SYPRO-Orange (Invitrogen, diluted 1000-fold of the manufacturer's stock). Fluorescence intensities were measured from 25 to 96 °C with a ramp rate of 1 °C/min. $T_m$ was determined by plotting the intensity as a function of temperature and fitting the curve to a Boltzmann equation. Final graphs were generated using GraphPad Prism. Assays were carried out in triplicate.

**Methylene blue activity assay**. The methylene blue colorimetric assay was used to determine cysteine desulfurase enzyme activity, by monitoring sulphide production[13,30]. The standard assay buffer consists of 50 mM HEPES pH 7.5, 200 mM NaCl, 10 mM DTT and supplemented with either 100 μM EDTA or 50 μM ZnCl₂. Individually purified proteins, when noted, consisted of NFS1-ISD11-ACP (SDA) at 0.5 μM, ISCU (U) at 2.5 μM, and Frataxin (F) at 2.5 μM. The isolated complex of SDAUF was at 0.5 μM, and any excess of ISCU or FXN would bring that protein to 2.5 μM. Reactions of 800 μL volume were mixed in a 1.5 mL black Eppendorf tube, and initiated by adding 100 μM L-Cysteine and placed in 37 °C

incubator for 10 min (with FXN) or 15 min (without FXN). Reactions were quenched with 100 μL of 30 mM FeCl₃ in 1.2 N HCl and 100 μL of 20 mM N,N-dimethyl-p-phenylenediamine (DMPD) in 7.2 N HCl. Reactions were subsequently incubated at 37 °C incubator for 20 min, followed by centrifugation to remove precipitate and absorbance measurement at 670 nm. Concentration of sulfide was calculated via calibration of a standard curve of Na₂S.

**Bio-Layer Interferometry**. BLI experiments were carried out using a 16-channel ForteBio Octet RED384 instrument equilibrated at 25 °C in BLI buffer (50 mM HEPES pH 7.5, 200 mM NaCl, 5% Glycerol, 2 mM TCEP, 5 mM DTT, 0.5 mg/mL BSA). Biotinylated WT ISCU ($U_b$) was combined with a purified batch of SDA and separated on Superdex 200 increase size exclusion column for isolation of the $SDAU_b$ complexes. Complexes were at 0.1 mg/mL and loaded to the streptavin coated sensors. The concentration for FXN used ranged from 500 μM to 1 nM. Measurements were taken in a programme consisting of a 90-second association step followed by a 60-second dissociation step, on a black 384-well bottom-tilted assay plate (ForteBio). Prior to association, the baseline was allowed to stabilize for 30 sec, and signal from the reference sensors was subtracted from measurements in the protein-loaded sensors. Dissociation constants ($K_d$) were determined through plots of response vs. [FXN], using the one site-specific binding fit in GraphPad Prism (GraphPad Software).

**Small angle X-ray scattering**. SAXS experiments for the SDAU and SDAUF complexes were performed at 0.99 Å wavelength Diamond Light Source at beamline B21 coupled to the Shodex KW403-4F size exclusion column (Harwell, UK) and equipped with Pilatus 2 M two-dimensional detector at 4.014 m distance from the sample, $0.005 < q < 0.4$ Å$^{-1}$ ($q = 4\pi \sin \theta/\lambda$, $2\theta$ is the scattering angle). The samples were in a buffer containing 300 mM NaCl, 25 mM Hepes 7.5, 1 mM TCEP 2 % Glycerol, 1% Sucrose and the measurements were performed at 20° C. The data were processed and analyzed with Scatter and the ATSAS program package[31]. Scatter was used to calculate the radius of gyration Rg and forward scattering I(0) via Guinier approximation and to derive the maximum particle dimension Dmax and P(r) function. The *ab initio* model was derived using DAMMIF[32]. 20 individual models were created, then overlaid and averaged using DAMAVER[33]. FoxS[34,35] server was used for comparison of theoretical and experimental data. SAXS data and parameters are summarized in Supplementary Table 4.

**LC-MS$^n$ analysis**. Mass measurement was carried out using Synapt G2 HDMS (Waters, Milford, MA) instrument equipped with Lockspray system, quadrupole mass analyzer, trap collision cell, and time-of-flight mass analyzer in tandem. Liquid chromatography was performed using an ACQUITY UPLC system with an Agilent PLRP-S column (1000 Å, 5 μm, 50 × 2.1 mm) at a flow rate of 0.30 ml/min. MS/MS measurement on the acyl-chain conjugated with 4′-PPT was accomplished by ejecting them from ACP in the ESI source region through in-source CID with elevated sample cone voltage ~ 60 V, and then fragmented by trap CID with trap potential ~ 30 V after selection in the quadrupole. The Synapt G2 mass spectrometer was externally calibrated using NaI solution and further lock-mass calibration using leucine enkephalin at $m/z$ 556.2771$^+$ was applied to every spectrum to achieve reliable ppm mass accuracy for both the intact protein ions and the 4′-PPT acyl chains fragment ions. Waters MassLynx software (Versions 4.1) was used for data acquisition and analysis, including calculations of predicted masses and simulation of isotopic distributions. The mass measurement errors (Δ) are the differences between the experimental and predicted values expressed in mDa, or when divided by the predicted mass in ppm units.

**Grid preparation and data acquisition.** 3.5 μL of 1.5 mg/ml purified SDAUF complex was applied to the glow-discharged Quantifoil Au R1.2/1.3 grid (Structure Probe), and subsequently vitrified using a Vitrobot Mark IV (FEI Company). In order to overcome an orientation bias, 0.067 % (w/v, final concentration) n-octyl-β -d-glucopyranoside (BOG, Anatrace) was added to the sample prior freezing. Cryo grids were loaded into a Titan Krios transmission electron microscope (ThermoFisher Scientific) operating at 300 keV with a Gatan K2 Summit direct electron detector. Images were recorded with SerialEM in super-resolution mode with a super resolution pixel size of 0.543 Å and a defocus range of 1.2 to 2.5 μm. Data were collected with a dose rate of 5 electrons per physical pixel per second, and images were recorded with a 10 s exposure and 250 ms subframes (40 total frames) corresponding to a total dose of 42 electrons per Å$^2$. All details corresponding to individual datasets are summarized in Supplementary Table 1.

**Electron microscopy data processing.** A total of 4,260 dose-fractioned movies were gain-corrected, 2 x binned (resulting in a pixel size of 1.086 Å), and beam-induced motion correction using MotionCor2[36] with the dose-weighting option. The SDAUF particles were automatically picked from the dose-weighted, motion corrected average images using Gautomatch. CTF parameters were determined by Gctf[37]. A total of 1,316,416 particles were then extracted using Relion 2.0[38] with a box size of 200 pixels. The 2D, 3D classification and refinement were performed with Relion 2.0. Two rounds of 2D classification and one round of 3D classification were performed to select the homogenous particles. After selecting particle coordinates, per-particle CTF estimation was refined using the program Gctf[37]. One set of 267,153 particles was then submitted to 3D auto-refinement with C2 symmetry imposed and resulted in a 3.2 Å map (Supplementary Figs. 4, 5). All 3D classifications and 3D refinements were started from a 60 Å low-pass filtered version of an ab initio map generated with VIPER[39]. To evaluate the contribution of imposed symmetry in the result, 3D refinement was repeated using the same set of 267,153 particles without imposing symmetry and produced a 3.4 Å map (Supplementary Fig. 4). Since the overall structures with/without imposing symmetry are nearly identical, the C2 symmetry density map was used for model building. All resolutions were estimated by applying a soft mask around the protein complex density and based on the gold-standard (two halves of data refined independently) FSC = 0.143 criterion. Prior to visualization, all density maps were sharpened by applying different negative temperature factors using automated procedures[40], along with the half maps, were used for model building. Local resolution was determined using ResMap[41] (Supplementary Fig. 4).

**Model building and refinement.** The initial template of the SDAUF complex was derived from a homology-based model calculated by SWISS-MODEL based on the crystal structures of (NFS1-ISD11-ACP-ISCU)$_2$ complex (PDB: 5WKP)[5] and human frataxin (PDB: 3S4M)[42] as the templates for NFS1, ISD11, ACP, ISCU, and FXN, respectively[43]. Each subunit was docked into the C2 symmetry full EM density map using Chimera[44] and followed by manually adjustment using COOT[45]. To prevent overfitting and confidently refine the ligands, the complete model with the ligands was subjected to global refinement and minimization in real space using the module phenix.real_space_refine in PHENIX[46] against one of EM half maps (working half map) with default parameters. The improvement of the model was monitored using the FSC curves of the complete model vs. the working half map (FSC$_{work}$) or the free half map (FSC$_{free}$). The complete model was subjected to an additional round of manually adjustment using COOT and followed by real-space refinement using PHENIX against the full EM map. The geometry parameters of the final models were validated in COOT and using MolProbity[47] and EMRinger[48]. These refinements were performed iteratively until no further improvements were observed. Residues at the distal ends of each chain were not built due to the poor densities. The local resolution map showed that the ACP subunits are more dynamic compared to the main body of the SDAUF complex (Supplementary Fig. 4c). The final refinement statistics were provided in Supplementary Table 1. Model overfitting was evaluated through its refinement against EM half maps. FSC curves were calculated between the resulting model and the working half map as well as between the resulting model and the free half and full maps for cross-validation (Supplementary Fig. 1). Figures were produced using PyMOL[49] and Chimera[44].

**Reporting summary.** Further information on research design is available in the Nature Research Reporting Summary linked to this article.

## Data availability

The cryo-EM data were deposited to the Protein Data Bank ('6NZU') and the EMDB ('EMD-0560', 'EMD-0561'). The raw data for Table 1, Fig. 1c, and Supplementary Figs. 1, 2, 3 and 7 are provided as a Source Data file. Other data are available from the corresponding authors upon reasonable request.

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

## Acknowledgements

The Structural Genomics Consortium is a registered charity (Number 1097737) that receives funds from AbbVie, Bayer Pharma AG, Boehringer Ingelheim, Canada Foundation for Innovation, Eshelman Institute for Innovation, Genome Canada, Innovative Medicines Initiative (EU/EFPIA) [ULTRA-DD grant no. 115766], Janssen, Merck & Co., Novartis Pharma AG, Ontario Ministry of Economic Development and Innovation, Pfizer, São Paulo Research Foundation-FAPESP, Takeda, and Wellcome Trust [092809/Z/10/Z]. N.G.F. and W.W.Y. are further supported by funding from the Pfizer Rare Disease Consortium.

## Author contributions

C.B., S.H., W.W.Y. designed the experiments. N.G.F., X.Y., X.F., H.J.B., A.M., J.F.N. and C.S-D. performed the experiments. N.G.F., S.H., and W.W.Y. wrote the manuscript.

## Additional information

**Competing interests:** Xiaodi Yu, Xidong Feng, Alain Martelli, Joseph F. Nabhan, Christine Bulawa, and Seungil Han are employees of Pfizer Inc. Nicolas G. Fox, Henry J. Bailey, Claire Strain-Damerell and Wyatt W. Yue declare no competing interests.

