## [Peer Review File · Nature Communications]

Reviewers' comments:

Reviewer #1 (Remarks to the Author):

The paper « Structure of the human frataxin-bound iron-sulfur cluster assembly complex provides insight into its activation mechanism » by Fox and collaborators reports the 3D structure, obtained by cryo-EM, of the assembly complex responsible for the biogenesis of iron-sulfur (Fe-S) clusters in mitochondria. This paper follows two recently published papers showing complexes without frataxin, which is an important protein in eucaryotes as demonstrated by the defects in human patients suffering from Friedreich ataxia. Thus, these data provide insight into the activation mechanism of this complex which was shown in vitro to rely on frataxin. In this complex, FXN binds at the interface of two NFS1 and one ISCU subunits, facilitating Fe-S cluster production by unlocking the zinc inhibition and stabilizing key loop conformations of NFS1 and ISCU at the protein-protein interfaces. These results represent an important contribution to the field as it helps understanding the molecular mechanisms underlying an essential cellular process.

Without being an expert of cryo-EM, the procedure used seems standard including for crossed data validation (calculation of half maps).

In the part 'Model building and refinement', « The initial template of the SDAUF complex was derived from a homology-based model calculated by SWISS-MODEL. », please give the templates (pdb codes) for each subunit with full reference information. Also the reference for using MolProbity should be provided.

In the remaining parts, I have the following comments :

1. Overall architecture of the SDAUF complex

Are the residues involved in electrostatic interactions between frataxin and NFS1 conserved in homologous sequences/structures ? A structural alignment showing residue conservation for both partners would help.

Furthermore, observation of the acidic ridge should refer to previous observation in 'Crystal structure of human frataxin' by Dhe-paganon et al, J. Biol. Chem. 2000, which should be cited.

2. FXN Interactions with NFS1 dimer interface and C-terminus

« Beyond the two extensive interfaces with FXN, NFS1 further contributes its C-terminal 20 aa (Ser437-His457) to wrap around the ISCU surface, with terminal residues Gln456-His457 anchored by FXN Asn151, Tyr175 and His177 via potential H-bonds (Supplementary Fig. 9). »

An additional panel to Supp figure 9, showing the electron density around this C-terminal end would be beneficial, as this region can be flexible.

3. FXN binds to two key regions on ISCU.

« This interaction, which may account for the weaker binding caused by the FXN(N151A) variant (Fig. 2b,d), is mediated by significant changes of the ISCU Ala-loop conformation in SDAUF as compared with SDAU-Zn²⁺ (rmsd ~6Å), and zinc-free SDAU (rmsd ~2Å) structures (Fig. 3a). » Please give the number of superposed alpha carbons used for the rmsd calculation.

4. On a general note, it may be worth correlating the interaction parameters of FXN variants with the SDAU complex with the clinical symptoms observed in human patients bearing these mutations if possible.

5. Conclusion

« This work also represents one of very few reported cryo-EM structures of <200 kDa and >3.5 Å resolution for both membrane and soluble proteins. »

The structure was determined at 3.2 Ang. resolution, so it should be <3.5 Å.

6. As mentioned in the conclusion, FDX is another important actor as it provides electrons for the

formation of the Fe-S cluster. It may be worth discussing better how this protein would cycle and integrate the cluster based on the current literature showing interactions with some of the assembly complex subunits.

Other minor points :

1. Page 4 : « ...acyl-chains are clearly the dominated species... » replace dominated by dominant or dominating.
2. Page 5 : « ...the conserved ISCU Ala-loop (Ala66-Asp71), contributing the conserved Cys69... », maybe change by « the conserved ISCU Ala-loop (Ala66-Asp71), containing the conserved Cys69 »
3. Page 5 : Maybe moderate the following statement by replacing « This explains why FXN alone cannot bind ISCU without NFS1 » by « This likely explains why FXN alone cannot bind ISCU without NFS1 »
4. Page 9 : « ...The concentration for FXN used ranged from 500 mM to 1 nM... ». This should be rather 500 μ M.

Nicolas Rouhier

Reviewer #2 (Remarks to the Author):

The manuscript by Fox et.al. describes the structure of iron-sulfur cluster assembly complex bound to its activator frataxin. This is the first structure of the activated complex. The complex contains four human and one E.coli subunit, which should not have influence on the main findings of this study however. Authors solved single particle cryo-EM structure of the complex which was further characterized and validated by mutagenesis and BLI measurements that together provide a very detailed picture of the interactions between iron-sulfur cluster assembly complex and frataxin. This is a significant advance in the field and this work would be of interest to structural biologists and biochemists working with protein complexes.

Before this paper can be published, the authors should address the following issues and questions.

- 1) From the introduction/discussion it is not very clear if Zn is bound to the complex in mitochondria and if its activation by frataxin is essential in vivo. Does frataxin play any additional role in FeS cluster biogenesis? What is the biological role of the regulation of iron-sulfur cluster assembly complex by frataxin. These questions would be asked by readers from different disciplines and fields when reading the paper.
- 2) In Figure 1b PLP is not visible.
- 3) In the discussion of FXN-NFS1 interaction it is first mentioned that the salt bridges are formed, but then in the text is stated that they can potentially form, which is confusing. Authors should clarify this paragraph and indicate the quality of the EM map in this region: is there a clear density for the side chains forming the salt bridges? At resolution of 3.2 Å the conformations of the side chains is often ambiguous, therefore observed H-bonds should be considered as plausible or potential H-bonds unless EM density is of very high quality.
- 4) Figure S1d – comparison of binding curves and corresponding Kd will be more clear if they are plotted in log scale on X axis.
- 5) Fig S3. Why SDA+U+F activity is much higher than that of SDA+U in presence of EDTA?
- 6) Table S1. It should be indicated in the table that the images were recorded in super resolution mode. Accuracy of resolution estimation should be rounded to the first decimal digit, i.e. 3.2 and 3.4 Å.
- 7) The refinement statistics of the model is surprisingly good, as is also indicated in PDB report. 0% of Ramachandran and side chain outliers is something I have not seen before. This may be a sign of an extremely good building and refinement but seems to be unusual. Authors should comment on this. Were all the atoms/residues in polypeptide chains of the complex modelled?
- 8) It is quite surprising that from 1.3 millions of particles after 3D classification only 270 thousand were left, which is around 20% only. While it is quite common, for proteins known to be very

fragile and unstable, here the reason is less obvious to the reviewer. I would appreciate comments from the authors on why it happens and whether 3D classes that were not used for the final high-resolution reconstruction show interpretable density that can provide information about properties of the protein complex.

9) Grid preparation: The concentration of BOG added to the sample before vitrification should be indicated.

10) Model refinement: Model refinement against half-map is not something I am aware of as being a common practice. If it is a validated method, a reference should be provided. Each half-map would need to be filtered separately to lower resolution to avoid refinement against noise, also each half-map has lower SNR meaning that final refinement will be not as good as if it is done against a complete map. It is also not obvious if the second half can be used as 'free' set for validation since it is used for FSC calculation which in turn is used for map filtering. Here references or more detailed explanation of the refinement process should be provided with justification of why it is a correct approach, alternatively the structure should be refined against the full map.

Reviewer #3 (Remarks to the Author):

Nature Communication: NCOMMS- 19-00003-T

Title: Structure of the human frataxin-bound iron-sulfur cluster assembly complex provides insight into its activation mechanism

Author(s): Nicholas G. Fox, Xiaodi Yu, Xidong Feng, Alain Martelli, Joseph F. Nabhan, Claire Strain-Damerell, Christine Bulawa, Wyatt W. Yue, Seungil Han

Comments to the Author

This paper reports the first structure of the FXN-bound active human complex, containing the hetero-pentamer (NFS1)₂-(ISD11)₂-(ACP)₂-(ISCU)₂-(FXN)₂ (SDAUF). This structure obtained by cryo-electron microscopy at 3.2 Å resolution delineates for the first time in any organism, the interactions of FXN with the component proteins. Two recent crystal structures depicting the inactive SDA and SDAU sub-complexes of the ISC biosynthesis machinery, lacking the activator FXN, have been determined. Even though these structures brought interesting information, the Fe-S biogenesis field awaited the active SDAUF complex structure! In addition to the detailed crystal structure of SDAUF, this work presents also some biophysical as well as biochemical studies using both wt and variants FXN. Therefore, the present work reveals crucial information which provide insights into frataxin activation mechanism and also offers an explanation of how FRDA clinical mutations can affect complex formation and therefore FXN activation.

The main inputs brought by the SDAUF complex structure are the following: FXN occupies a cavity at the interface of both NFS1 and one ISCU subunits. FXN binds with NFS1 dimer interface and NFS1 C-terminus. Concerning ISCU interaction, compared to the SDAU/SDAU-Zn₂₊ structures published before there is significant displacement of ISCU, up to 2.0 Å away from the core, leading to modifications of the ISCU local environment. FXN binds to two key regions on ISCU. One is through the conserved ISCU Ala-loop (Ala66-Asp71), containing the conserved Cys69, required for Fe-S biosynthesis, that interacts with FXN Asn151 as well as with the Zn₂₊ coordinating ligand Asp71. The other region is through the conserved ISCU L131PPVKLHCSM140 sequence motif, containing the 'L131PPVK135' sequences recognized by the GRP75/HSCB chaperones, the Cys138 (sulfur acceptor) and the Met140, the residue reportedly determining if Fe-S biosynthesis is FXN-dependent (as in eukaryotes) or not (prokaryotes). Interestingly, Met140 packs against FXN Pro163, while the M140I substitution (present in SDAU/SDAU-Zn₂₊ structures) sterically clashes with Pro163, unless the 163 position adopts the bacterial equivalent amino acid, Gly (ecoli Gly68). Hence, the present structure illustrates the evolutionarily distinct Met:Pro pairing in eukaryotes and Ile:Gly in prokaryotes. Finally, the Zn₂₊ environment on ISCU, thought to mimic the Fe-S cluster, is different in SDAUF complex compared to that one in SDAU complex. In the reported SDAU-Zn₂₊ structure, ISCU Asp71, Cys95 and His137, and NFS1 Cys381 form the Zn₂₊ ligation. In the present SDAUF-Zn₂₊ structure, zinc is ligated by ISCU Asp71, Cys95 and Cys138. This rearranged metal coordination frees up ISCU His137 to interact with FXN Trp155, Leu156, Pro163, and ISCU Lys135. Most importantly, NFS1 Cys381 is also freed from Zn₂₊ ligation, now available

for sulfur transfer. Therefore this structure elucidates how FXN binding to SDAU complex causes significant conformational changes to ISCU and primes its key regions (Ala-loop and LPPVK region) to facilitate mobility of NFS1 Cys-loop for sulfide formation and transfer during Fe-S assembly. To conclude, this is an outstanding work. I consider the structure of SDAUF-Zn²⁺ complex as a very significant piece of scientific work. It provides the framework for future mechanistic studies on the dynamics of SDAUF complex during Fe-S assembly. This work should be published as fast as possible in Nature Communication.

Reviewer 1

In the part 'Model building and refinement', « The initial template of the SDAUF complex was derived from a homology-based model calculated by SWISS-MODEL. », please give the templates (pdb codes) for each subunit with full reference information.

Response – The templates (PDB codes: 5WKP and 3S4M) with reference information (ref #5,43) have been included in our revised manuscript, page 12.

Also the reference for using MolProbity should be provided.

Response – The reference for MolProbity has been included in our manuscript (ref #48).

1. Overall architecture of the SDAUF complex

Are the residues involved in electrostatic interactions between frataxin and NFS1 conserved in homologous sequences/structures? A structural alignment showing residue conservation for both partners would help. Furthermore, observation of the acidic ridge should refer to previous observation in 'Crystal structure of human frataxin' by Dhe-paganon et al, J. Biol. Chem. 2000, which should be cited.

Response – The majority of residues involved in the frataxin-NFS1 are conserved among orthologues, as illustrated by a new structure-based sequence alignment figure incorporated as new Supplementary Figure S9. As suggested by the reviewer, we have also cited the reference (Dhe-paganon 2000; ref #21) on page 5.

2. FXN Interactions with NFS1 dimer interface and C-terminus

«Beyond the two extensive interfaces with FXN, NFS1 further contributes its C-terminal 20 aa (Ser437-His457) to wrap around the ISCU surface, with terminal residues Gln456-His457 anchored by FXN Asn151, Tyr175 and His177 via potential H-bonds (Supplementary Fig. 9).» An additional panel to Supp figure 9, showing the electron density around this C-terminal end would be beneficial, as this region can be flexible.

Response – We thank the reviewer for the suggestion. We have incorporated an additional panel to the now Supplementary Figure S10, showing the density around the C-terminal of NFS1.

3. FXN binds to two key regions on ISCU.

« This interaction, which may account for the weaker binding caused by the FXN(N151A) variant (Fig. 2b,d), is mediated by significant changes of the ISCU Ala-loop conformation in SDAUF as compared with SDAU-Zn²⁺ (rmsd ~6Å), and zinc-free SDAU (rmsd ~2Å) structures (Fig. 3a). »

Please give the number of superposed alpha carbons used for the rmsd calculation.

Response – The number of superposed C^α atoms is now provided for the above sentence on pages 5-6, as well as for another sentence on page 4 that refers to rmsd calculation.

4. On a general note, it may be worth correlating the interaction parameters of FXN variants with the SDAU complex with the clinical symptoms observed in human patients bearing these mutations if possible.

Response – We have provided sentences on page 6 to discuss how our BLI interaction data for the three FRDA-causing variants (N146K, W155R, R165C) correlate with clinical severity.

5. Conclusion

« This work also represents one of very few reported cryo-EM structures of <200 kDa and >3.5 Å resolution for both membrane and soluble proteins. »

The structure was determined at 3.2 Ang. resolution, so it should be <3.5 Å.

Response – The typo has now been fixed on page 7.

6. As mentioned in the conclusion, FDX is another important actor as it provides electrons for the formation of the Fe-S cluster. It may be worth discussing better how this protein would cycle and integrate the cluster based on the current literature showing interactions with some of the assembly complex subunits.

Response – We have provided sentences and a new reference in the conclusion paragraph (pages 7-8) to briefly describe how FDX could possibly engage with the SDAUF complex, with a disclaimer that future studies are much needed to further understand the mechanism.

Other minor points :

1. Page 4 : « ...acyl-chains are clearly the dominated species... » replace dominated by dominant or dominating.

2. Page 5 : « ...the conserved ISCU Ala-loop (Ala66-Asp71), contributing the conserved Cys69... », maybe change by « the conserved ISCU Ala-loop (Ala66-Asp71), containing the conserved Cys69 »

3. Page 5 : Maybe moderate the following statement by replacing « This explains why FXN alone cannot bind ISCU without NFS1 » by « This likely explains why FXN alone cannot bind ISCU without NFS1 »

4. Page 9 : « ...The concentration for FXN used ranged from 500 mM to 1 nM... ». This should be rather 500 μM.

Response – We thank the reviewer for the suggested changes – they have now been incorporated.

Reviewer #2 (Remarks to the Author):

1) From the introduction/discussion it is not very clear if Zn is bound to the complex in mitochondria and if its activation by frataxin is essential in vivo. Does frataxin play any additional role in FeS cluster biogenesis? What is the biological role of the regulation of iron-sulfur cluster assembly complex by

frataxin. These questions would be asked by readers from different disciplines and fields when reading the paper.

Response – The binding of Zn^{2+} towards components of the complex (e.g. ISCU) and its effect on complex activity (e.g NFS1 desulfurase) have so far been observed *in vitro*. The *in vivo* relevance of the Zn^{2+} contribution remains to be determined. Frataxin is essential as indicated by human genetics (FRDA) and mouse knockout studies. Additional roles have been speculated for frataxin, beyond that of an allosteric activator, such as providing/delivering the iron source, although further evidence is much needed. We have rewritten sentences in the introduction (Page 3) to clarify the above.

2) In Figure 1b PLP is not visible.

Response – A modified Figure 1b was provided with clear views on the ligands (PLP, 8Q1, ZN).

3) In the discussion of FXN-NFS1 interaction it is first mentioned that the salt bridges are formed, but then in the text is stated that they can potentially form, which is confusing. Authors should clarify this paragraph and indicate the quality of the EM map in this region: is there a clear density for the side chains forming the salt bridges? At resolution of 3.2 Å the conformations of the side chains is often ambiguous, therefore observed H-bonds should be considered as plausible or potential H-bonds unless EM density is of very high quality.

Response – We agree with the reviewer that at 3.2 Å resolution, interpretation of electron density for H-bond and salt bridges may not be straightforward. We have now taken this into consideration, and have described all H-bonds and salt bridges as potential interactions throughout the text.

4) Figure S1d – comparison of binding curves and corresponding K_d will be more clear if they are plotted in log scale on X axis.

Response – We have provided an additional plot of the K_d binding curves with log scale on X axis (as new Figure S1e).

5) Fig S3. Why SDA+U+F activity is much higher than that of SDA+U in presence of EDTA?

Response – The data from Figure S3 show that both SDA+U+F (five-way complex formed by reconstitution of recombinant proteins) and SDAUF (five-way complex formed by co-expression) have significantly higher NFS1 activity than SDA+U, with and without Zn^{2+} , because of the activation of frataxin. It is well established that SD, SDA, and SDAU complexes have only basal activity.

It is of note that the five-way complexes from reconstitution or co-expression exhibit some dissociation equilibrium for the frataxin protein, hence rendering NFS1 mobile loop cysteine to be susceptible to act as a coordinating ligand for the zinc that is bound to ISCU causing inhibition of the enzyme. We presented the body of data from Figure S3, to illustrate that we can prepare a stable form of the five-way complex (reducing frataxin dissociation) by further supplementation of frataxin to the complex (SDAUF+F). We attributed the intactness of frataxin in the SDAUF+F sample to the lack of observed Zn^{2+} inhibition, because of the conformational changes FXN bestows upon the complex. We have rewritten sentences on page 4 to explain the above better.

6) Table S1. It should be indicated in the table that the images were recorded in super resolution mode. Accuracy of resolution estimation should be rounded to the first decimal digit, i.e. 3.2 and 3.4 Å.

Response – The estimated resolutions in Table S1 have been updated and rounded to the first decimal as per reviewer’s suggestion.

7) The refinement statistics of the model is surprisingly good, as is also indicated in PDB report. 0% of Ramachandran and side chain outliers is something I have not seen before. This may be a sign of an extremely good building and refinement but seems to be unusual. Authors should comment on this. Were all the atoms/residues in polypeptide chains of the complex modelled?

Response – We much appreciate the above comments from the reviewer. The quality of the EM maps allowed us to build/refine the model confidently and accurately. Several residues with alternative conformations were also observed from the density map. We feel that the current and other published, high resolution EM maps have good phase information to achieve better geometry compared to x-ray density maps. Similar examples can be found from PDB codes: 5K0Z, 5K12, 6NBD, and 6CVM, etc.

The following statement was added to further clarify the quality of current model in the model building and refinement section, page 12: *“Residues at the distal ends of each chain were not built due to the poor densities. The local resolution map showed that the APC subunits are more dynamic compared to the main body of the SDAUF complex (Supplementary Fig. S4c).”* Taken together, the EM map results in a model with the Ramachandran values as shown in Table S1 (0.00% outlier, 7.27% Allowed, 92.73% Favored).

8) It is quite surprising that from 1.3 millions of particles after 3D classification only 270 thousand were left, which is around 20% only. While it is quite common, for proteins known to be very fragile and unstable, here the reason is less obvious to the reviewer. I would appreciate comments from the authors on why it happens and whether 3D classes that were not used for the final high-resolution reconstruction show interpretable density that can provide information about properties of the protein complex.

Response – In order to overcome the orientation bias, not only n-octyl-β -d-glucopyranoside (BOG) was added into the sample prior freezing, we also intended to pick more particles during the auto-picking step by lowering the threshold. In this case, we had higher chance to pick up the particles with the rare orientations. Nevertheless, the junk particles were also more likely to be included during the auto-picking step. The junk particles were mostly removed during multiple rounds of the 2D classifications.

The 3D classification steps further resulted in the reduction in the number of particles. The SDAUF complex was composed of 10 polypeptide chains. The dynamic nature of the protein complex causes structural heterogeneity, which limits the high-resolution structure determination. In order to deal with the structural heterogeneity, we further performed multiple rounds of 3D classifications to screen the best particles. One class with 267,153 particles was selected for final 3D refinement and reconstruction. The other classes, either missing parts of densities or being less populated, did not result in any reconstructions that were suitable for interpretation confidently. No alternative conformations of SDAUF complex showed up during the 3D classification steps, which indicates that the structure we determined represents the most favorable state under the current experimental conditions.

9) Grid preparation: The concentration of BOG added to the sample before vitrification should be indicated.

Response – The concentration of BOG (0.067 % w/v, final concentration) has now been included in our manuscript, page 11.

10) Model refinement: Model refinement against half-map is not something I am aware of as being a common practice. If it is a validated method, a reference should be provided. Each half-map would need to be filtered separately to lower resolution to avoid refinement against noise, also each half-map has lower SNR meaning that final refinement will be not as good as if it is done against a complete map. It is also not obvious if the second half can be used as ‘free’ set for validation since it is used for FSC calculation which in turn is used for map filtering. Here references or more detailed explanation of the refinement process should be provided with justification of why it is a correct approach, alternatively the structure should be refined against the full map.

Response – Following the reviewer’s suggestion, a more detailed description of the model building and validation strategies have been included in our revised manuscript, page 12.

We did not low-passed filter the half maps for the following two reasons: first, even the half maps have relatively high SNRs (the noise signal disappeared with slightly increase of the contour level); second, low-passed filtering the maps would decrease the features on the ligand parts used for modeling properly.

Reviewer #3 (Remarks to the Author):

To conclude, this is an outstanding work. I consider the structure of SDAUF-Zn²⁺ complex as a very significant piece of scientific work. It provides the framework for future mechanistic studies on the dynamics of SDAUF complex during Fe-S assembly. This work should be published as fast as possible in Nature Communications.

Response – We thank Reviewer #3 for the very positive remarks on our work.

REVIEWERS' COMMENTS:

Reviewer #2 (Remarks to the Author):

The authors addressed all my comments in a satisfactory way. In my opinion the paper can be published in its current form.